# HyClear: A Novel Tissue Clearing Solution for One-Step Clearing of Microtissues

**DOI:** 10.3390/cells11233854

**Published:** 2022-11-30

**Authors:** S. Soroush Nasseri, Erika M. J. Siren, Jayachandran N. Kizhakkedathu, Karen Cheung

**Affiliations:** 1School of Biomedical Engineering, University of British Columbia, Vancouver, BC V6T 1Z3, Canada; 2Centre for Blood Research, University of British Columbia, Vancouver, BC V6T 1Z3, Canada; 3Department of Chemistry, University of British Columbia, Vancouver, BC V6T 1Z1, Canada; 4Department of Pathology and Laboratory Medicine, University of British Columbia, Vancouver, BC V6T 2B5, Canada; 5Department of Electrical and Computer Engineering, University of British Columbia, Vancouver, BC V6T 1Z4, Canada

**Keywords:** tissue clearing, spheroid, organoid, microscopy, 3-D, HPG, DMSO

## Abstract

3-D cell cultures are being increasingly used as in vitro models are capable of better mimicry of in vivo tissues, particularly in drug screenings where mass transfer limitations can affect the cancer biology and response to drugs. Three-dimensional microscopy techniques, such as confocal and multiphoton microscopy, have been used to elucidate data from 3-D cell cultures and whole organs, but their reach inside the 3-D tissues is restrained by the light scattering of the tissues, limiting their effective reach to 100–200 µm, which is simply not enough. Tissue clearing protocols, developed mostly for larger specimens usually involve multiple steps of viscous liquid submersion, and are not easily adaptable for much smaller spheroids and organoids. In this work, we have developed a novel tissue clearing solution tailored for small spheroids and organoids. Our tissue clearing protocol, called HyClear, uses a mixture of DMSO, HPG and urea to allow for one-step tissue clearing of spheroids and organoids, and is compatible with high-throughput screening studies due to its speed and simplicity. We have shown that our tissue clearing agent is superior to many of the commonly used tissue clearing agents and allows for elucidating better quality data from drug screening experiments.

## 1. Introduction

In vitro 3-D cultures have emerged as a better model of in vivo tissues compared to monolayers. They result in a better mimicry of the in vivo tissues as they allow for the cells to maintain stroma-cell interactions that control cell signalling, differentiation, and development [1,2,3]. They can also generate mass transfer gradients similar to in vivo, which can result in the accumulation of metabolites or depletion of oxygen and nutrients, which can in turn change the cell metabolism and response to external stimuli (e.g., treatment with anticancer agents) [4,5,6,7]. This can provide great opportunities for more efficient drug screenings, especially when many promising drug candidates in in vitro studies fail in further stages of animal and clinical tests, thus wasting a huge amount of time, effort and capital.

The great promises of 3-D cell cultures, however, are impeded by the how difficult elucidating data from 3-D cell cultures can be, especially from the cells in the inner cores of the cultures where different responses to external stimuli is more likely to occur. The traditional methods involve the fixation of the tissues in membranes, physical sectioning using microtomes and imaging each slice. This a process too laborious and time consuming to be adapted in a high throughput drug screening processes [8]. 3-D microscopy techniques, such as confocal and multi-photon microscopy techniques, would provide a better alternative [9], but they are limited by the optical properties of the tissues. Thicker tissues scatter the light passing through, resulting in poor z-stack resolution with increasingly dimmer cores. This leaves the more interesting part of the spheroids, their inner core, for evaluation. This is mostly due to scattering at the lipid (cell-membrane)-aqueous interface due to the refractive index (RI) mismatch between the lipid-rich tissue and the surrounding aqueous solutions [10,11]. While the signal attenuation might address some of these challenges by modulating the laser intensity through the depth [12], the images will still be blurry due to scattering. An example of such phenomenon is shown in Figure 1, and imaged using a two-photon microscope (Olympus FV1000 MPE, as detailed in Section 2.3.1), which depicts the gradual loss of signal and increased blurring with the increase of depths in an optical section. Since some of the interesting phenomena with relevance to drug screening (such as necrotic cores) begin to form in spheroids with a 400–600 µm diameter [13], the lack of imaging access to those areas can defeat at least some of the purposes of using 3-D cell cultures.

Several protocols and techniques (collectively called “tissue clearing”) have been developed to address the problem of non-uniform intensity and blurring in 3-D microscopy. The main reason these problems occur is the refractive index (RI) mismatch between the solid parts of the tissues (RI ~1.45) and the water-based solutions (RI ~1.33) surrounding them [10,14,15,16]. This results in scattering of the light passing through and the subsequent dimming and blurring happening, especially at the centre of the spherical tissues in deeper z-stack images, where compared to the edges, light must pass through more tissue to reach the objective lens [17,18]. Thus, an effective strategy to overcome the problem of scattering within tissues would be to reduce the refractive index (RI) disparity within the tissues, by substituting the surrounding liquid with a “clearing agent” liquid of higher refractive index [19,20,21]. An alternative approach could be to alter the optical properties of the solid parts of the tissues such that the RI is reduced to something closer the surrounding water-based solutions [22,23]. Additionally, other methods have been proposed in which the scattering lipids have been completely removed after embedding the tissue in a hydrogel [15].

Over the years, research groups have introduced many tissue clearing protocols, which are mostly developed for larger tissues and involve steps such as solvent change, etc. These are not easily implementable in smaller tissues, such as tumour spheroids, while taking a long time to perform in their original format [14] and causing morphological changes [24], which makes them inconvenient or sometimes impossible to use for small tissues. We have previously demonstrated an on-chip implementation of several important methods described by other groups for clearing spheroids [25]. While the on-chip implementation of clearing protocols can be beneficial for imaging spheroids in smaller experiments, it is not a reliable method for imaging the large volume of samples as required in high-throughput drug screening methods, which usually relies on familiar formats, such as well plates to both grow and image the samples, and on liquid handling robots to manage the 3-D cultures, such as spheroids, while conducting the drug screening experiment. For a tissue clearing protocol to be usable for small 3-D cell cultures like spheroids, it should rely on a rapid, single-step application of a clearing agent, which would be manageable for liquid handling robots that is the current method for handling experiments at an industrial scale. Moreover, a good tissue clearing agent for microtissues should be compatible with rather sensitive fluorescent proteins used in many cell-based experiments and should preferably introduce minimal auto-fluorescence and tissue shrinkage.

Here, we present a novel tissue clearing agent (HyClear) using DMSO and hyperbranched polyglycerols (HPGs) to be used in the high throughput clearing of a large number of microtissues. By reducing the mismatch between the tissue and the surrounding liquid, the application of this clearing agent can reduce the light scattering and absorption with increasing imaging depth, making imaging at higher depths possible with higher signal intensity and less blurring. HyClear permits non-invasive endpoint imaging of whole spheroids and other microtissues using a low-viscosity solution applied in a one-step process that could be readily used in microfluidic systems and automated high-throughput multi-well systems alike. In addition, it also permits the preservation of fluorescent proteins, sustains morphological features of the microtissues, results in none to minimal autofluorescence in microtissues, and leads to higher or comparable clearing of tissues in comparison to multistep clearing processes that are adaptable for spheroids and other microtissues.

## 2. Materials and Methods

### 2.1. Cell Culture and Spheroid Formation

#### 2.1.1. Cancer Cell Culture

The NCI-H1299 lung cancer cells already transfected with GFP (ATCC), and the NCI-HFL-1 lung cancer fibroblast cells (ATCC) in DMEM/F12 (Sigma-Aldrich, St. Louis, MO, USA) that was supplemented with 10% fetal bovine serum (Sigma-Aldrich), 2.5 mM L-glutamine and Anti-Anti (Invitrogen, Carlsbad, CA, USA). Cells were passaged upon reaching confluence and were lifted off the surface by 0.25% Trypsin/EDTA (Invitrogen). The H1299 and HFL-1 cells were kept in DMEM/F12 media (Corning, Corning, NY, USA) with 10% Fetal Bovine Serum (Sigma-Aldrich) and Anti-Anti (Gibco, Grand Island, NY, USA). The HUVEC cells (ATCC, Manassas, VA, USA) were cultured in EBM-2 (Lonza, Basel, Switzerland) media supplemented with an EGM-2 bullet kit (Lonza). Standard cell culture procedures were followed for the maintenance of the cells. Cells were incubated in a humid atmosphere with 5% CO_2_ and 37 °C.

#### 2.1.2. Cell Line Generation

For the screening study that was used here as an example, we used NCI-H1299 epithelial lung cancer cell line NCI-HFL-1, lung fibroblasts and HUVEC cells. To identify the cells visually without the need for subsequent staining, we made fluorescent cell lines of these cells. We acquired lentiviral plasmids (pLV-mCherry and pLV-eGFP) from Addgene. We chose the lentiviral transfection of our cells to obtain stable fluorescent cell lines.

The plasmids were multiplied using the prescribed protocols of the MiniPrep and MaxiPrep kits. Then, 7.5 million HEK 293T (ATCC) cells per 10 cm plate were seeded in cell culture petri dishes in standard grow media (DMEM + 10% FBS + 1% Pen-Strep) and were grown overnight to 90% confluence. On the day after, for each cell culture dish, we mixed the prepared DNA solution (500 µL OptiMEM + 20 µg total DNA) with the Lipofectamine solution (500 µL OptiMEM + 40 µL Lipofectamine 3000 (2 µL/µg DNA), mixed and incubated for 5 min at room temperature) and incubated it for 20 min at room temperature. The mixture was then added to the cell culture plates and incubated for 4–6 h in the incubator. The culture was then rinsed with PBS. Afterwards, 6 mL of a media mixture containing Opti-MEM reduced serum media (Gibco 31985070) + 5% FBS + 1% non-essential amino acid (NEAA) + 1% Na-pyruvate was added to the cells. After 24 h, the supernatant was collected, and another fresh 6 mL of the same media was added again and collected after an additional 16 h. The lentivirus in the supernatant was then concentrated using Lenti-X™ concentrator using the manufacturer protocol, was aliquoted in 20–50 µL volumes in cryotubes, flash-frozen in liquid nitrogen and kept in −80 °C.

To infect the cells using the produced viruses, we trypsinized the intended cells from a T25 flask, centrifuged and resuspended them in 4 mL of DMEM with 8 µL/mL polybrene. Then, we divided the mixture into 4 wells of a 6-well plate (1 mL per well). Add 30 µL of the intended lentivirus solution to each well, except for a well kept as control. The 6-well plate was then spun at 3200× *g* at 25 °C for 5 min, and then for 1 h at 2500× *g* at 25 °C. Afterwards, we rinsed the virus with wash media twice and added 2 mL of the respective growth media to each well. The cultures were monitored for growth and infection rate after a day and transferred to a bigger culture vessel when near confluent. After reaching the desirable number, the cells were sorted using the BD Influx Cell Sorter (at UBC Flow Cytometry Facility) machine for a further selection of the fluorescent cells. The sorted cells were plated again and frozen later for future use.

#### 2.1.3. Tumour Spheroid Formation

For the mono-culture spheroids, H1299 cells were detached from the surface, counted using a hemocytometer, and diluted to the desired concentration so that a certain number of cells is plated in each well. 200 µL of the cell solution was pipetted into each well of an ultralow attachment 96-well plate (Corning, CAT# 4515) such that it contains 1000–2500 cells per well. The initial number of cells was selected depending on the desired size and harvest date of the spheroids. The spheroids formed within 96 h. Media had to be changed if they were to be kept for more than a week.

For tri-culture spheroids, we trypsinized and counted the cells, and diluted them such that there were approximately 500–2000 cells in each 200 µL of the cell solution going inside each well. The exact number of those cells is detailed in Section 3.4. For co-cultures, the different cell types were mixed from the beginning. The settling of the cells in the bottom can be helped with a brief spinning using a centrifuge. If the cell cultures required a different media, a 1:1 mixture of their respective media was used. The spheroids formed after almost 4 days. If the spheroids were intended to be kept for more than a week, 50 µL of the media can be added to the wells to feed the cells.

#### 2.1.4. Spheroid Fixation

Spheroids were fixed using 2% para-formaldehyde (Ted Pella, Redding, CA, USA) in PBS at room temperature overnight. To fix the spheroids in the 96-well plates, an amount of 4% PFA, equal to that of the supernatant media in each well was added to each well, which would bring the PFA concentration to the desired 2%.

### 2.2. Tissue Clearing

#### 2.2.1. SeeDB Tissue Clearing

Fructose (Bio Basic, Markham, ON, Canada) solutions of 20%, 40%, 60%. 80%, 100% and 115% wt/vol fructose were dissolved in ultrapure water, after which α-thioglycerol was added to them to reach a final concentration of 0.5%. All solutions were made at room temperature, except the 100% and 115% that were dissolved at 65 °C and then cooled down prior to the addition of α-thioglycerol [19].

#### 2.2.2. Clear^T2^ Tissue Clearing

A 40% polyethylene glycol (PEG) solution in ultrapure water was prepared by dissolving PEG 8000 (Bio Basic, Markham, ON, Canada) in warm water. 50% formamide/20% PEG 8000 and 25% formamide/10% PEG solutions were made by mixing formamide with the previously made solution and water [20].

#### 2.2.3. ScaleSQ(0) Tissue Clearing

We chose the ScaleSQ(0) solution among the ScaleS protocols for its rapid clearing and absence of detergents. A solution of 9.1 M urea and 22.5% (*w*/*v*) sorbitol in ultrapure water was prepared by dissolving urea (Anachemia, Richmond, BC, USA) and sorbitol (Bio Basic) in warm water [23]. The solution was always kept above 30 °C during the experiment by using hot packs and lens warmers to prevent precipitation of the highly concentrated urea.

#### 2.2.4. HyClear Tissue Clearing

Different HPG formulations with molecular weights were prepared to the concentrations used in the experiments. The HPG solutions were mixed with DMSO and PBS to the desired concentration and applied through a microfluidic chip (when compared to other clearing protocols) or added to 96-well plates containing spheroids. HPGs (HPG-1K (Mn-1000 Da) and HPG 3 K (Mn-3000 Da) were synthesized in the Kizhakkedathu laboratory following protocols described in recent publications [26,27].

#### 2.2.5. Tissue Clearing Setup

After fixing the spheroids, the fixative was removed and they were washed carefully with PBS in the original well plate and loaded into glass-bottom 96-well plates for imaging. All the pipetting was done using wide-bore pipette tips to prevent damaging spheroids during transfer to the glass-bottom plate, and while washing the spheroids. The spheroids were imaged while in PBS, then the PBS was carefully removed and HyClear clearing solutions were added to the desired wells. The plate was spun briefly to bring the spheroids to the bottom of the wells, after which the spheroids were imaged again after waiting for at least 20 min.

For clearing whole organs, the fixed mouse organs were immersed overnight in the HyClear solution on a rocking shaker.

### 2.3. Imaging

#### 2.3.1. Microscopy

Imaging was conducted using two microscopes based on availability and access, but care was taken to compare images only within the same system and settings:Olympus FV1000 MPE microscope and excited by a MaiTai DeepSee Ti:Sapphire laser, with a 25× water dipping objective lens optimized for TPM (XLPLN25XWMP) with an NA of 1.05 and a working distance of 2 mm for all imaging, combined with 495–540 nm (green) and 576–630 nm (red) filters.Zeiss LSM 880 AxioObserver microscope with Plan-Apochromat 20×/0.8 M27 and N-Achroplan 10×/0.25 Ph1 M27_b lenses, with the laser set at 405, 488 and 594 nm. The spheroids were transferred to a 96-well plate with flat cover glass #17 before imaging. Unless specified, all the images are taken using this microscope.

#### 2.3.2. Image Processing

The following protocol was used for the image processing as the spheroids were large and often did not fit within the boundaries of the images. A circular region in the centre of the spheroid was chosen in each z-stack image, in which the average signal intensity was measured and corrected versus the background (using Fiji). The corrected signal intensities were used to calculate the average intensity as a measure of the overall brightness of the fluorescence in the whole stack and each depth throughout the whole stack and in each depth. Then, we analyzed the signals within the spheroid region. We also calculated the standard deviation within the spheroid region, which gives a measure of the blurriness of the image detected within the spheroid region, with a wider range of intensities indicative of less blurriness and higher contrast. This is like how contrast-detection autofocus systems work, that is, by trying to maximize the standard deviation of the detected image through the focus range.

## 3. Results

### 3.1. Mixture of DMSO and HPG Solution Increases Fluorescence Imaging Penetration Depth

To develop a clearing solution with the characteristics suitable for clearing large arrays of spheroids and other 3-D micro-cultures, we started by using dimethyl sulfoxide (DMSO). It has a refraction index of 1.48 and viscosity of 1.99 cP at 25 °C (water has a 0.890 cP and common honey varieties range between ~2000–23,000 cP [28]). DMSO has also been investigated as a clearing agent for clearing skin samples for bright-field microscopy, with possible topical application in human patients [29,30]. Given its high refractive index and relatively low viscosity DMSO can be a good candidate for use in clearing solutions. Our preliminary applications of DMSO for tissue clearing showed that pure DMSO quenches the fluorescence of proteins. This was mostly due to the removal of the water necessary for protein fluorescing [14], which has been found to stabilize the excited state of some proteins [31]. We found that using 75% DMSO in PBS (RI = 1.45) could help with the quenching of the fluorescence of proteins.

Rawat et al. have reported that polyethylene glycol (PEG) solutions can have a stabilizing effect on proteins, due to the binding of PEG molecules with proteins [32]. In fact, this has been used for reducing the fluorescence quenching in the ClearT tissue clearing protocol. We investigated other polymers that can have the same effect on protein stabilization, thus preserving fluorescent signals in proteins when in contact with DMSO, while having a high refractive index at a relatively low concentration and viscosity. To this end, we tested PEG molecules of different molecular weights (MW) as well as other similar polymers, such as methoxy-PEG, polyvinyl alcohol (PVA), and hyperbranched polyglycerols (HPG), checked different stable mixtures of them to stabilize the fluorophores while maintaining strong clearing capability.

An interesting class of polymers that we found to have similar effects with PEG in preventing fluorescent protein quenching were hyperbranched polyglycerols (HPG). HPGs are a class of ultra-compact hydrophilic polymers with globular structures that have a 50–65% dendrimeric structure, are chemically stable and have low intrinsic viscosity [26]. Compared to the linear PEG molecules, an extremely high concentration of HPG solutions could be made, while also keeping the solutions less viscous and free-flowing.

We came up with the prototype of HyClear by mixing a 1.56 g/mL HPG 1 K solution in PBS (the concentration at which the HPG solution had an RI = 1.49) with the 75% (*v*/*v*) DMSO solution we tried before at a 1:1 ratio. Figure 2 compares the z-stack images of the spheroids obtained through TPM at equivalents depths (different due to moderate shrinkage of the spheroids after clearing), before and after clearing using the perfusion of the 0.78 g/mL HPG 1 K + 37.5% DMSO solution (hereafter referred to as HyClear-Pre1) in a microfluidic device, as was done in our previous work [25]. The perfusion of the clearing solution resulted in the clearing of the samples in less than 5 min, increasing the fluorescence signal in the samples by more than 4-fold on average in depths higher than 100 µm.

The HyClear-Pre1 solution was compared to the other clearing protocols of its class, using a microfluidic chip, as implemented in our previous study [25]. The clearing solution presented in this work results in mostly a higher increase of fluorescence signal on average (Figure 3a) and through the depth of the spheroids (Figure 3b), as compared to the protocols we adapted in our previous work. While the clearing performance of the SeeDB protocol among all the others is closer to our protocol, it should be noted that SeeDB relies on several steps and works with very high viscous solution, which makes SeeDB a slow (<1 h) and complicated clearing process for microtissues, even in our on-chip adaptation of it [25]. However, the HyClear-Pre1 solution uses a free-flowing low-viscosity solution in a one-step process that can be completed in less than 5 min, making it well-suited to use for clearing spheroids and other microtissues in both lab-on-a-chip and in high-throughput multi-well plate systems.

### 3.2. Optimizing the DMSO and HPG Mixture Tissue Clearing Protocol

While we found that the aforementioned DMSO and HPG 1 K mixture works well in clearing microtissues compared to the other common tissue clearing techniques, we wanted to further investigate whether we can improve its efficiency by changing the properties of HPG, such as its MW, and also try to find out whether the shrinking of tissues after clearing can be reduced without compromising the tissue clearing capability. To this end, we have tested several modifications in a design of experiments that considered many different modifications to the 0.78 g/mL HPG 1 K + 37.5% DMSO formula. The tests were done on spheroids made with NCI-H1299 cells and was fixed, as described earlier. There were five–six spheroids for any condition tested.

A slightly lower amount of DMSO (30% vs. 37.5%) has been used in the experiments described in this section. Unlike the microfluidic chips used in the previous section and in [25], we cannot completely remove the existing liquid in the well-plates containing the spheroids since it might result in the collapse of the larger 3-D tissues (20 µL of liquid was left to avoid that). This would inadvertently result in the undesirable subsequent dilution of the clearing solutions. To overcome that, we tried adding 80 µL of a 1.25× concentrated clearing solution that would become the 1× desired solution once added to the well. However, we found that having DMSO at a higher concentration than 37.5% *v*/*v* will result in precipitations. Thus, we had to keep the DMSO at 37.5% *v*/*v*, which resulted in a 30% *v*/*v* solution upon addition to the wells.

In the first modification to our formula, we investigated using other variants of HPG with higher molecular weight (HPG 1 K vs. HPG 3 K). The compositions can be found in Table 1.

Figure 4a shows the change of intensity in clearing solutions made with different molecular weights of HPG. The results show that HPG 1K and 3K work very similarly in terms of how much they increase the intensity of fluorescence before and after tissue clearing (Figure 4a), with very similar tissue shrinkage profiles (Figure 4b). HPG 3 K has a higher viscosity than HPG 1K in solutions of a similar concentration, and HyClear-Pre2 (with HPG 3K) did not result in any significant improvement over HyClear-Pre1, thus showing that we should continue using HPG 1K in our formula and we should try to optimize it with other additives to address the considerable amount of tissue shrinkage in the HyClear-Pre1 results.

In order to address the problem of tissue shrinkage, we investigated the use of urea to hyper-hydrate the membranes of the tissue and thus prevent tissue shrinkage, as has been described before [22,23]. The tested formulations can be found in Table 1. The maximum concentration of urea we tested was 6.4 M, as we found that higher concentrations resulted in precipitation. The addition of urea decreases the shrinkage significantly (Figure 5), but where the addition of urea is especially beneficial, is at higher depths where the clearing capability of the clearing solutions is increased dramatically with the addition of urea (Figure 6). This might be due to the less dense tissue that light passes through, due to less shrinkage caused by hyper-hydration of the tissues in the presence of urea. All the different formulae resulted in a less blurry image after clearing, as measured by the increase in the variance of the images before and after clearing (Figure 7). Combining all the factors, we have demonstrated the clearing mixture HyClear-Pre5 (RI = 1.47, µ = 12.7 cP) as the optimal solution of choice for tissue clearing and we call it HyClear.

### 3.3. Clearing Spheroids with HyClear

We have tested the HyClear on different types of samples to assess its efficacy in tissue clearing. Figure 8 depicts an example of a large spheroid made of GFP producing NCI-H1299 lung cancer cells, cleared using the HyClear and imaged using two-photon microscopy (Olympus FV1000 MPE microscope with a 25× water immersion objective). Figure 8c,d are histogram-adjusted versions of (a) and (b), maximizing the visibility of the signals in each depth. It is evident from the comparison of (a) and (b) that HyClear increases the fluorescence signal in each depth. From the comparison of (c) and (d), we can see how HyClear can help to visualize larger spheroids of about 500 µm, resulting in sharper images in each depth.

### 3.4. HyClear Incorporated in Screening Experiments

As mentioned earlier, the main reason behind developing HyClear was to have a tissue clearing solution that can be fitted in large-scale screening experimental workflows that use 3-D cell cultures like spheroids, so that we could realize their full potential as a tissue model by being able to visualize the cells deep inside them. Hereby, we are demonstrating an example of fitting HyClear into a screening experiment involving several spheroids in a 96-well plate format.

The aim of this experiment is to establish a more sophisticated 3-D culture model that can mimic the in vivo tissues better, by including other cell types that normally exist in tissues, such as fibroblasts and endothelial cells. These new models can be especially used for modelling diseases like lung cancer, where anti-angiogenesis have been used with some lung cancers such as non-small-cell lung cancer [33,34], where the addition of cells, such as endothelial cells, with the potential for vasculature formation will make the spheroids a potentially useful model for testing with anti-angiogenesis drugs. Whereas a simple monoculture of lung epithelial cells would not be capturing the sophisticated nature of the lung tissues, developing methods for producing coculture lung spheroids could potentially compete with or complete the picture illustrated by other sophisticated models, such as lung-on-a-chip devices.

In this experiment, we wanted to produce lung co-culture models with epithelial, fibroblast and endothelial cells. We aim to have a model in which all different cell types can thrive, but we would particularly like to see the more sensitive endothelial cells thrive in the coculture. Since the growth rates of the different cell types are different, starting from the same number of cells results in the complete take-over of the spheroids by the fastest-growing cell type. Other researchers who have made similar co-cultures from these cell types have come up with some rules of thumb regarding the ratio of the different cell types. For example, Lazarri et al. have suggested a 9:1:4 ratio for the (fibroblast: epithelial: endothelial) ratios [35]. We tried different ratios, as described in Table 2, where the number of fibroblasts was kept the same, and some different ratios of epithelial to endothelial cells were looked at from conditions A to D, and with half of those numbers from conditions E to H. We were looking for the growth of cell types other than the epithelial cells, and specifically, whether we can find any considerable growth of HUVEC cells inside the coculture. We used the HyClear as an important part of the experiment to find out if such vascular structures form.

To promote the growth of HUVEC cells, EBM-2 media has some special growth factors, among which vascular endothelial growth factor (VEGF) is of considerbale importance. Since EBM-2 endothelial growth media is diluted when mixed with the required media for epithelial and fibroblast cells (DMEM/F12), we supplemented the media with additional VEGF so that its level comes back to the concentration present in EBM-2 (here called VEGF-1X). To see the effect of VEGF, we also made spheroids of every condition with 2- and 3-times the VEGF concentration as that of VEGF-1X (called VEGF-2X and 3X, respectively).

To visualize the cells, the H1299 cells were tagged with GFP, and the HUVEC cells were transfected to express mCherry. To discern the HFL-1 fibroblast cells, we incubated the spheroids with CellTracker Blue (Invitrogen), such that any cell that is only blue is a fibroblast, and others with colocalized GFP and mCherry are NCI-H1299 and HUVEC cells, respectively.

We had 24 conditions, for which we made three spheroids for each condition, for a total of 72 spheroids. Obtaining data for this number of spheroids would be very difficult if we rely on the conventional tissue clearing protocols, and would be virtually impossible while doing more sophisticated experiments with a much higher number of samples common in industrial drug screening practice. However, HyClear provides a practical and effective way to overcome this problem. To perform the tissue clearing in this experiment, we moved the spheroids to a cover-glass bottom plate using wide pipette tips (to prevent any damage to the spheroids). All the buffer in each well was removed, except 20 µL in each well to keep the spheroids from dismantling. Then, 80 µL of HyClear was added to the wells, and the plate was spun using a centrifuge briefly to make sure all the spheroids settle down in the bottom. The imaging was started at about 30 min after the addition of HyClear. To make the imaging faster, all spheroid locations were identified first so that the whole imaging process could be done in one turn without the need for manual intervention.

Figure 9 shows the maximum Z-projection images of spheroids from each condition in the experimental design. The blue cells are the NCI-HFL-1 fibroblasts, the ones that are green or cyan are the epithelial NCI-H1299 cells and the ones that are red or orange denote the HUVEC cells. These images could provide us some clues about which condition has resulted in better growth of HUVEC cells or any vascular formation, but they are not conclusive as any vascular formation should show up in the 3-D reconstruction of each image. As the number of images is quite high for the processing-intensive 3-D reconstruction and visual inspection, we designed another method to find the spheroids with the highest probability of having endothelial cell growth without initial visual inspection. We segmented the spheroids based on the ubiquitous CellTracker in the blue channel, to find out the area of the spheroids. Then, we created a Sum Projection Image of the red channel and measured its mean and standard deviation in the segmented area. The red pixels in that area could either be from autofluorescence or the HUVEC cells. If they were caused by the HUVEC cell fluorescence, we would expect them to be closer together with higher intensity than autofluorescence pixels, hence resulting in a higher standard deviation. To normalize the standard deviation to the total intensity of the images (different among individual spheroids), we divided the standard deviation by the average intensity of the red pixels in the segmented area, also known as its relative standard deviation, or the coefficient of variance (C.V.).

We checked the spheroids with a C.V. larger than 0.9, and among them found that some significant growth of HUVEC cells have formed within the spheroids (depicted in Figure 10) made using condition G, with ratios of 10:1:5 at the VEGF-3X condition. While we only intended to demonstrate the process of incorporating HyClear in large-scale experiments here, we should add that these results are, to the best of our knowledge, the first reports of coculture lung spheroids.

The spheroid shown in Figure 10 is an example from condition F, from which we found several cases of HUVEC growth and demonstrate notably that by using HyClear we were able to see HUVEC-mCherry cells deep in the core of spheroids. Figure 10 also shows how HyClear can be used in multiplexing applications, conserving both fluorescent proteins and stains at the same time without interference with each other.

Overall, this example demonstrates how we could use HyClear in experiments where we desire to obtain data from the core of 3-D tissues in experiments with a large number of samples. Due to the simplicity of the processes, which involves a simple aspiration of the supernatant media and the addition of the low-viscosity HyClear solution (µ = 12.7 cP), they could all easily be implemented in robotic fluid handling systems. HyClear can also handle the multiplexing of fluorescent dyes and proteins, as was demonstrated in this example.

### 3.5. HyClear for Clearing Whole Tissues

While HyClear was formulated with spheroids and microtissues in large experimental setups in mind, we also looked to see how it would perform in clearing tissues larger than spheroids such as whole mouse organs. The protocol remained the same (A single immersion step in the HyClear Reagent). We used transgenic mouse muscles with fibroblasts that are fluorescently labelled with EGFP. We cleared the tibialis anterior (TA) muscle and the heart muscle in this demonstration of the HyClear. As evident from Figure 11 and Figure 12, the HyClear increased the imageable depth of the tissues (with imaging parameters that were kept the same throughout the experiment for each sample type).

The imageable depth for the mouse heart in Figure 11 increased from about 50 µm to about 155 µm (~3× increase). In the mouse TA tissue (Figure 12), the imageable death increased from about 160 µm to 350 µm.

## 4. Discussion

We have demonstrated a novel tissue clearing protocol “HyClear” with excellent tissue clearing capabilities. HyClear has low viscosity and is capable of one-step and high-throughput clearing of 3-D microtissues, such as spheroids, which is readily applicable in workflows involving 96 and 384-well plates handled with liquid handling robots. HyClear does not alter morphological features, as it induces minimal tissue shrinkage and negligible induced autofluorescence. It is also compatible with fluorescent proteins and fluorescent stains and can be used in assays involving multiplexing fluorophores. This will make it easier to use tissue clearing for imaging 3-D cell cultures as a part of an image-based drug screening platform, facilitate data-acquisition from 3-D cultures in high-content drug discovery experiments usually performed using 2-D cultures, building on the tools developed for 2-D cultures while taking advantage of the data that 3-D cultures can provide by imaging not only their surfaces, but also their core. As the cells in the core of the spheroids facing nutrient and oxygen limitation can respond differently to drugs, the use of the tools described here may result in important findings on the efficacy of drugs in the environment simulating the interior of tumours where the cells are restricted of nutrients and oxygen. In addition to microtissues, we have also tested HyClear on tough-to-clear mouse whole organs to find that it can clearly increase the imageable depth by more than two-fold.

Tissue clearing has evolved considerably over the last century to improve the data acquisition from intact tissues. While there are multiple methods available for different types of tissues, no single method can claim to be superior in every situation. HyClear has been developed with the needs of imaging microtissues, such as organoids and spheroids in mind, and delivers an easy-to-use and effective method for clearing large numbers of microtissues rapidly. Nonetheless, we have also shown that HyClear can be used in larger tissues as well, especially where more sophisticated methods are not available. In future, HyClear can be used as the basis for establishing multistep clearing protocols for clearing larger tissues even more effectively. Additionally, for antibody labeling of endogenous proteins, we expect that the immunohistochemical labeling will occur after fixation and before clearing. HyClear has been shown to conserve the intracellular fluorescent proteins, so we expect it should be compatible with other proteins, such as antibodies as well. This will be confirmed in future work.

## 5. Patents

University of British Columbia has submitted a provisional patent application based on the research described in this manuscript with S.S.N., E.M.J.S., J.N.K. and K.C. as inventors.

## Figures and Tables

**Figure 1 cells-11-03854-f001:**
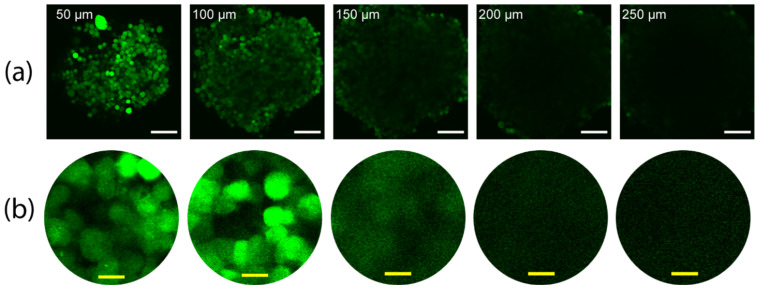
The gradual dimming and blurring in images of a z-stack with depth, taken using two-photon microscopy (**a**) z-stack images of a spheroid made with EGFP-positive NCI-H1299 lung cancer cells. Labels show the depth of each image in the spheroid. (**b**) 5× zoomed images of the center of the images in row a and histogram-adjusted for better visibility in the higher depths. White and yellow scale bars denote 100 µm and 10 µm, respectively.

**Figure 2 cells-11-03854-f002:**
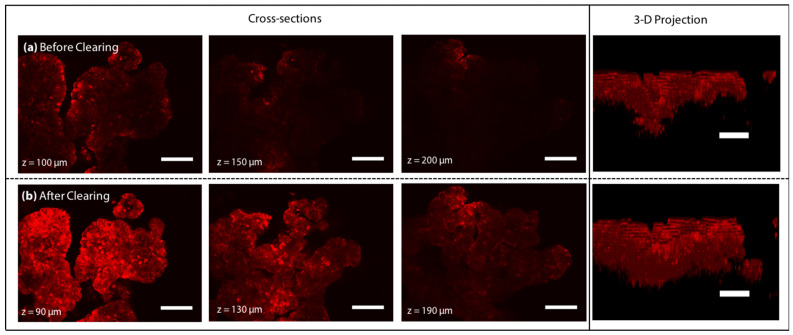
Two-photon microscopy image slices of a tumour spheroid at 100, 150 and 200 µm deep into the tissue (**a**) before clearing with the corresponding z-stacks and (**b**) after clearing with HyClear-Pre1 solution (changed due to shrinkage). The 3-D projection image of the spheroids is shown on the right. The scale bars represent 100 µm.

**Figure 3 cells-11-03854-f003:**
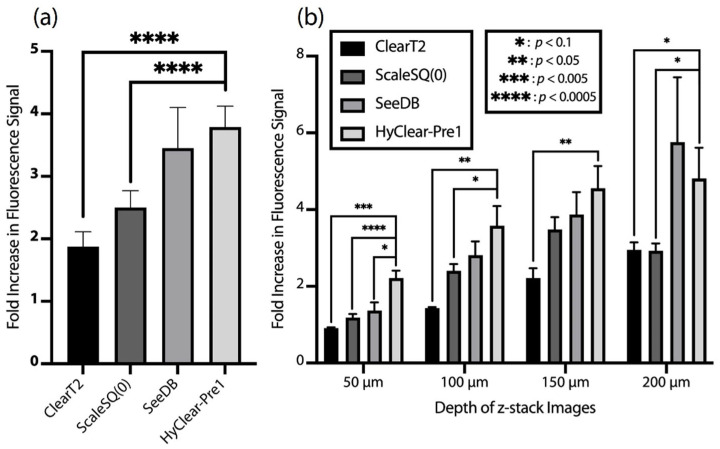
(**a**) The average overall fold increase of the fluorescence signal in the centre of the spheroids (in depths between 50 to 200 µm) in different clearing methods adapted for microtissues [3] versus HyClear-Pre1. (**b**) The average fold increase in the fluorescence signal after clearing in different imaging depths using some of the common clearing methods versus HyClear-Pre1. The significances are calculated using a heteroscedastic two-tailed *t*-test. The error bars represent the standard error of the mean.

**Figure 4 cells-11-03854-f004:**
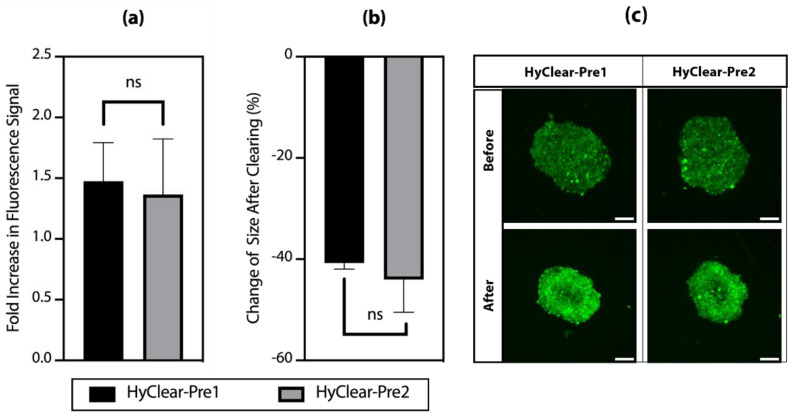
(**a**) The average fold increase in the fluorescence signal after clearing using different types of HPG. (**b**) The average change of the area of the cross-sectional images before and after clearing using different types of HPG. (**c**) An example of the MIP image of the spheroids before and after tissue clearing using different types of HPG. The error bars represent the standard error of the mean of five to six replicates.

**Figure 5 cells-11-03854-f005:**
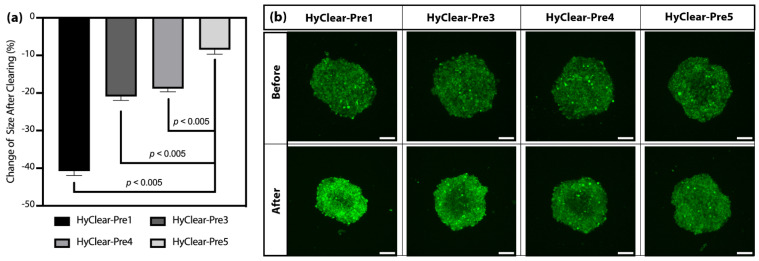
(**a**) The average change of the cross-sectional area of the cross-sectional images before and after clearing using HyClear with different concentrations of urea. (**b**) An example of the MIP image of the spheroids before and after tissue clearing using different concentrations of urea. The error bars represent the standard error of the mean of five to six replicates. The *p*-values are calculated using a heteroscedastic two-tailed *t*-test. The scale bars represent 200 µm.

**Figure 6 cells-11-03854-f006:**
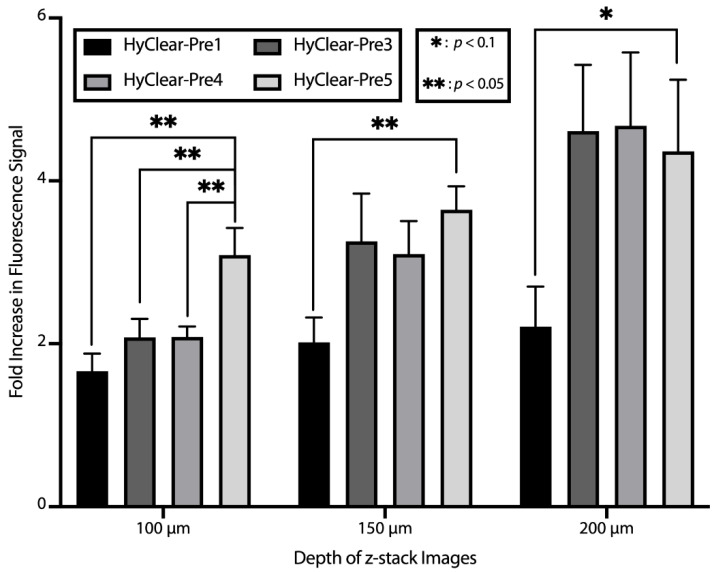
The average fold increase in the fluorescence signal after clearing in different imaging depths using different clearing methods versus HyClear. The *p*-values are calculated using a heteroscedastic two-tailed *t*-test. The error bars represent the standard error of the mean of five–six replicates.

**Figure 7 cells-11-03854-f007:**
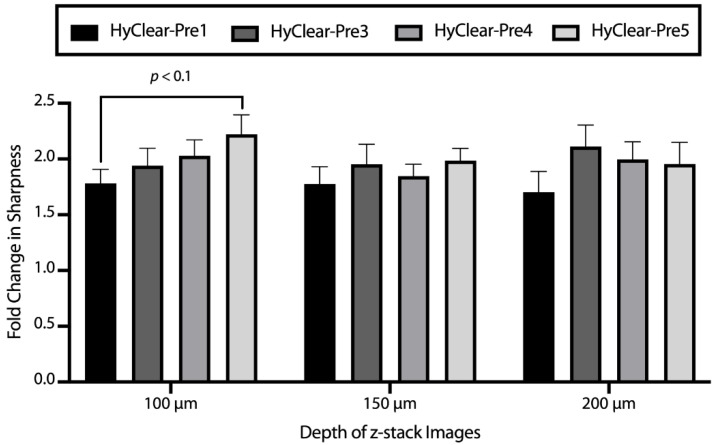
The average fold increase in image sharpness after clearing in different imaging depths using various HyClear formulations. All variations of HyClear improve the sharpness by more than ~1.6 times. The addition of urea suggests improving the sharpness, although the significance cannot be established in all cases. The *p*-values are calculated using a heteroscedastic two-tailed *t*-test. The error bars represent the standard error of the mean of five to six replicates.

**Figure 8 cells-11-03854-f008:**
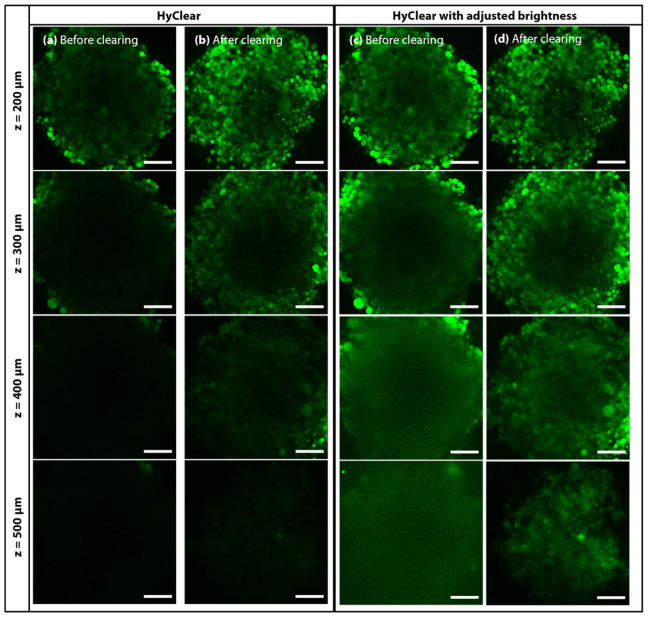
Two-photon microscopy image slices of a tumour spheroid at 200, 300, 400 and 500 µm deep into the tissue (**a**) before HyClear with 6.4 M Urea clearing with the corresponding z-stacks and (**b**) after clearing. The 3-D projection image of the spheroids is shown on the bottom. (**c**,**d**) are the histogram adjusted version of (**a**,**b**) at each depth, respectively. The scale bars represent 100 µm. The images were obtained using an Olympus FV1000 MPE microscope with a 25× water dipping objective.

**Figure 9 cells-11-03854-f009:**
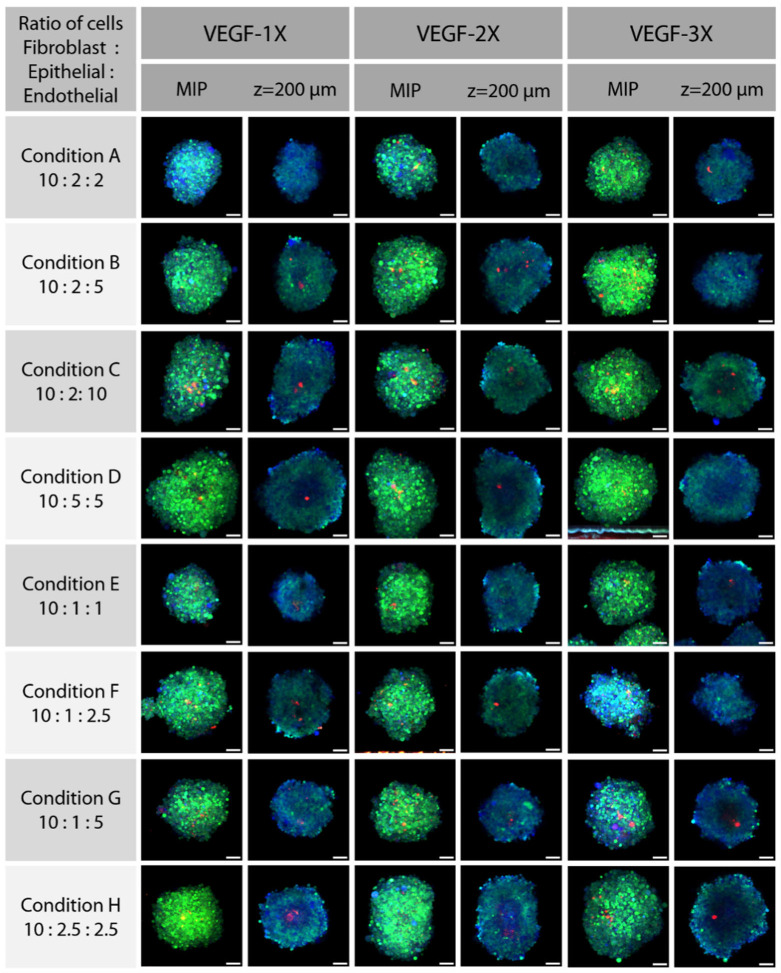
MIP and z-slice (at approximately 2/3 of the whole depth, at z = ~200 µm, histogram-adjusted) images of the spheroids at conditions described in Table 2, with different VEGF levels. Each whole spheroid is treated with CellTracker Blue; NCI_HFL-1 fibroblast cells were not fluorescent themselves, so they were only blue due to the CellTracker stain. NCI-H1299 cells are fluorescent with GFP, so they show as green to cyan. The HUVEC cells are brightly colored with mCherry, and show as red. The scale bars represent 100 µm.

**Figure 10 cells-11-03854-f010:**
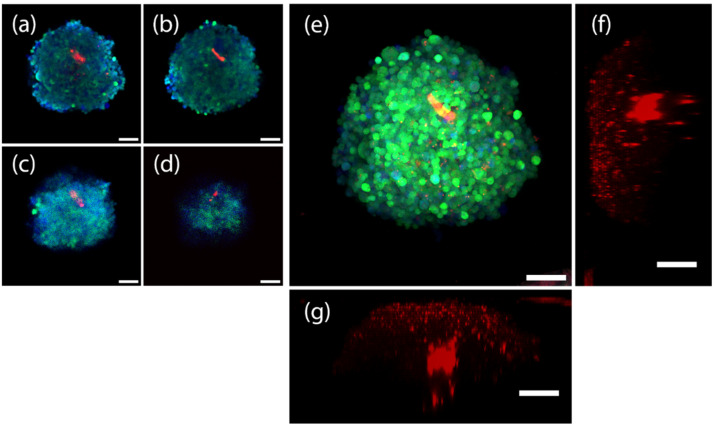
10× confocal images (after clearing with HyClear, histogram-adjusted) of a triculture spheroid of epithelial, fibroblast and endothelial cells, depicted by dGFP (green), CellTracker Blue (blue) and mCherry (red) colours, respectively. (**a**–**d**) are histogram adjusted z-slices of the spheroid at (**a**) 150 µm, (**b**) 200 µm, (**c**) 250 µm, and (**d**) 300 µm. (**e**) depicts the maximum projection image of the spheroid. (**f**,**g**) depict the 3D projection of the spheroid (red channel only) from the Y and X axes, respectively. Scale bars represent 100 µm.

**Figure 11 cells-11-03854-f011:**
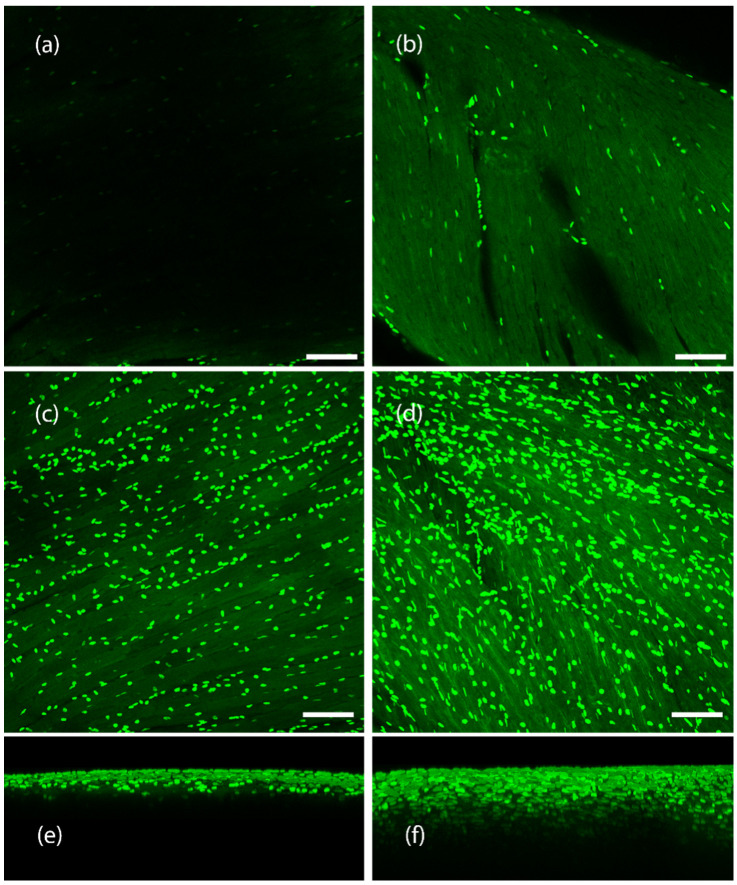
Clearing of a mouse heart tissue using HyClear. (**a**,**b**) Image slices of a mouse heart tissue with GFP-positive fibroblasts, using a 20× lens, at 44 µm, (**a**) before and (**b**) after tissue clearing. (**c**,**d**). MIP images of a mouse heart tissue using a 20× lens, with GFP-positive fibroblasts, (**c**) before and (**d**) after tissue clearing. 3-D reconstructed z-stack image of mouse heart before tissue clearing. (**e**,**f**) 3-D reconstructed z-stack image of mouse heart (**e**) before and (**f**) after tissue clearing. The imageable depth of the tissue increases from about 50 µm in image (**e**) (before) to about 155 µm in image (**f**) (after). The scale bars represent 100 µm.

**Figure 12 cells-11-03854-f012:**
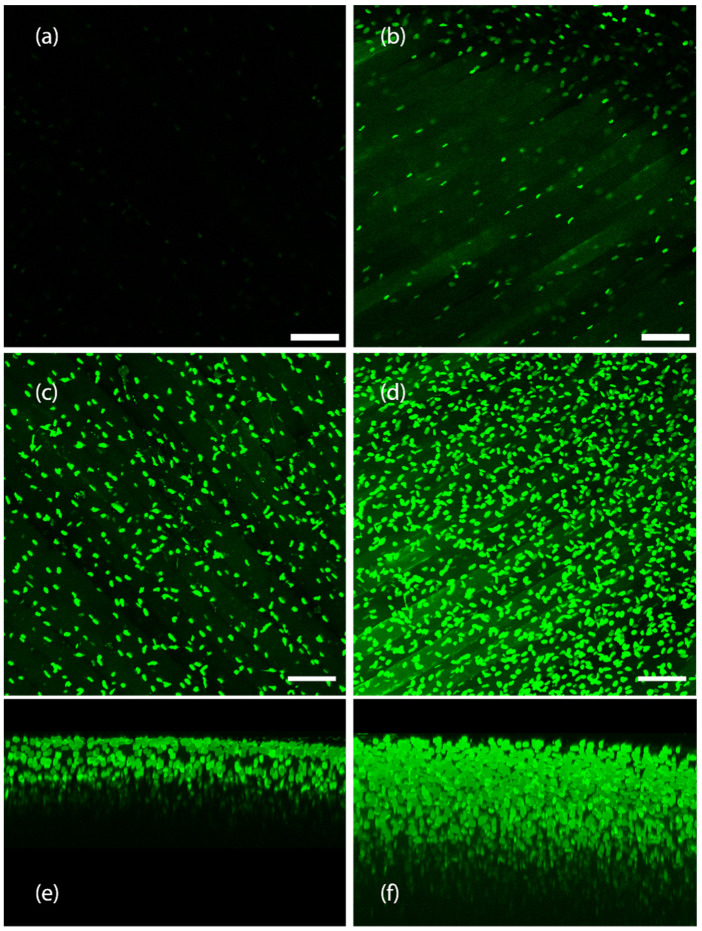
Clearing of a mouse TA muscle tissue using HyClear. (**a**,**b**) Image slices of a mouse TA muscle using a 20× lens with GFP-positive fibroblasts, using a 20× lens, at 150 µm, (**a**) before and (**b**) after tissue clearing. (**c**,**d**) MIP image of a mouse TA muscle using a 20× lens, with GFP-positive fibroblasts, (**c**) before and (**d**) after tissue clearing. (**e**,**f**) 3-D reconstructed z-stack image of mouse TA muscle € before, and (**f**) tissue clearing. The imageable depth of the tissue increases from about 160 µm in image (**b**) (before) to about 350 µm in image (**d**) (after). The scale 16 bars represent 100 µm.

**Table 1 cells-11-03854-t001:** The different formulations tested for optimizing HyClear.

Solution	HPG Concentration (g/mL)	HPG Molecular Weight (Da)	DMSO Concentration (*v*/*v*)	Urea Concentration (mol/L)
HyClear-Pre1	0.78	1K	30%	0
HyClear-Pre2	0.78	3K	30%	0
HyClear-Pre3	0.78	1K	30%	1.6
HyClear-Pre4	0.78	1K	30%	3.2
HyClear-Pre5	0.78	1K	30%	6.4

**Table 2 cells-11-03854-t002:** Experimental design to find the optimal starting point for the lung co-culture spheroids.

Condition	No. of HFL-1 Cells (Ratio)	No. of H1299 Cells (Ratio)	No. of HUVEC Cells (Ratio)	DMEM/F12 (µL)	EBM-2 (µL)
A	2500 (10)	500 (2)	500 (2)	100	100
B	2500 (10)	500 (2)	1250 (5)	100	100
C	2500 (10)	500 (2)	2500 (10)	100	100
D	2500 (10)	1250 (5)	1250 (5)	100	100
E	2500 (10)	250 (1)	250 (1)	100	100
F	2500 (10)	250 (1)	625 (2.5)	100	100
G	2500 (10)	250 (1)	1250 (5)	100	100
H	2500 (10)	625 (2.5)	625 (2.5)	100	100

## Data Availability

Not applicable.

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
