# Peer review of "HyClear: A Novel Tissue Clearing Solution for One-Step Clearing of Microtissues"

_cells, 2022, doi:10.3390/cells11233854_

Round 1

Reviewer 1 Report

In this manuscript by Nasseri et al, the authors describe the development and utility of HyClear, a tissue clearing formulation that is easily applied to 3D cultures and tissue sections to improve imaging quality at higher depths. Significant improvements over existing tissue clearing approaches are described, and the methods and approach is sound and clearly laid out. Data presented are convincing and mostly supports the conclusions made by the authors. However, I have the following comments that need to be addressed before publications. I suggest one experiment to better convince the utility of HyClear to improve imaging of spheroid cores, as well as other comments to improve data presentation.

Line 56, 273 etc: the PDF version of the manuscript seemed to have replaced “Figures” with an error message.

Figure 3a please indicate from how many independent experiments these graphs are derived from.

Figure 3b please include and describe statistical analysis.

Figure 7: it is not clear to me whether the different formulations are improving the sharpness of the images. Please include and describe statistical analysis and indicate which formulation was best for this purpose.

Figure 8: image in (b) and (d) at z=500um does not appear to be the same image adjusted for brightness, looking at the overall shape and the one bright cell.

Figure 10. The core of the spheroid in this image is approximately 120um from the surface, according to the scale provided. Studies in Figure 8 demonstrates that while HyClear does provide improved clarity at up to 200um depths, significant improvements in brightness and sharpness is more evident at higher depths. Therefore, the image provided here may not be the best representation of improvements made by HyClear because the core is likely to be seen well regardless of tissue clearance. I suggest imaging larger spheroids with core epithelial cells located >200um from the surface, with and without HyClear, to more convincingly demonstrate the utility of this procedure in 3D spheroid applications, as described.

Reviewer 2 Report

This manuscript describes a method using a new tissue clearing agent based on DMSO and hyper-branched polyglycerols. The protocol is adapted to small 3D structures as spheroids and organoids. The method is well written and results appear to be promising but some information for the protocols and more precise images need to be performed.

1-    A precise protocol scheme using spheroids and tissue clearing method would really help scientists to perform this type of experiment. Also, a precision about the type of 96 wells plates used to performed spheroids (to not adhere on the substrate), how the authors precisely and technically fix them, transfer them etcs.

As example, in point 2.1.4. is there a removal of the medium before fixation ?

In point 2.2.5 how the spheroids are washed? etcs…)…. For these reasons, a protocol scheme will really help).

2-    Could the authors explain on Figure 9, the authors should provide not only a maximum projection of the spheroids (where it is difficult to see the different type of cells and if it is possible to image the center of the spheroids),  also at least for some on the conditions, a confocal  plane of the center of the spheroids.

3-    Same than point 2, for Figure 11, a confocal plan image (at different z)  would help to see the “clarity” of the in vivo samples (which is difficult to see with a MIP)

4-    Could this clearing agent be compatible with endogenous protein labelling with antibodies (uDISCO, Clarity etcs) ?

5-    For the observed shrinkage of the spheroids, even minimal, is it homogeneous (conservation of AR ratio of the cells, shape) before and after the clearing?

Reviewer 3 Report

In the manuscript 'HyClear: A Novel Tissue Clearing Solution for One-Step Clearings of Microtissues' authors developed a method to enhance tissue clearing using DMSO, HPG, and urea. Despite the good attempt, manuscript requires minor revision.

Comments.

1.       How is HyClear-Pre better than seeDB?

2.       Figure 4 does not show any difference. How many times was it repeated? Can you show the comparative images?

3.       Figure 5 looks interesting but requires visible differences in images. Can you add images?

4.       In figure 6, surprisingly HyClear-Pre1,3 and 4 seem insignificant in thin sections, but significant in thick sections. How do you explain this?

Round 2

Reviewer 1 Report

All concerns have been addressed.

Reviewer 2 Report

The reviewer is happy with this new version with the modifications.